# Sleep Disorder and Cocaine Abuse Impact Purine and Pyrimidine Nucleotide Metabolic Signatures

**DOI:** 10.3390/metabo12090869

**Published:** 2022-09-15

**Authors:** Mayur Doke, Jay P. McLaughlin, Hamid Baniasadi, Thangavel Samikkannu

**Affiliations:** 1Department of Pharmaceutical Sciences, Irma Lerma Rangel School of Pharmacy, Texas A&M University Health Sciences Centers, College Station, TX 77843, USA; 2Department of Pharmacodynamics, College of Pharmacy, University of Florida, Gainesville, FL 32611, USA; 3Department of Biochemistry, University of Texas Southwestern Medical Center, Dallas, TX 75390, USA

**Keywords:** circadian rhythm, sleep disruption, cocaine abuse, metabolism, metabolomics, bioinformatics

## Abstract

Disturbances in the circadian rhythm alter the normal sleep-wake cycle, which increases vulnerability to drug abuse. Drug abuse can disrupt several homeostatic processes regulated by the circadian rhythm and influence addiction paradigms, including cravings for cocaine. The relationship between circadian rhythm and cocaine abuse is complex and bidirectional, and disruption impacts both brain function and metabolic profiles. Therefore, elucidating the impact of circadian rhythm changes and cocaine abuse on the human metabolome may provide new insights into identifying potential biomarkers. We examine the effect of cocaine administration with and without circadian rhythm sleep disruption (CRSD) on metabolite levels and compare these to healthy controls in an in vivo study. A metabolomics analysis is performed on the control, CRSD, cocaine, and CRSD with cocaine groups. Plasma metabolite concentrations are analyzed using a liquid chromatography electrochemical array platform. We identify 242 known metabolites compared to the control; 26 in the CRSD with cocaine group, 4 in the CRSD group, and 22 in the cocaine group are significantly differentially expressed. Intriguingly, in the CRSD with cocaine treatment group, the expression levels of uridine monophosphate (*p* < 0.008), adenosine 5′-diphosphate (*p* < 0.044), and inosine (*p* < 0.019) are significantly altered compared with those in the cocaine group. In summary, alterations in purine and pyrimidine metabolism provide clues regarding changes in the energy profile and metabolic pathways associated with chronic exposure to cocaine and CRSD.

## 1. Introduction

Circadian rhythm is an internal clock that is essential in maintaining day and night cycles and plays a role in the day-to-day physiological, behavioral, and metabolic regulation functions. This internal clock exists in the superchiasmatic nucleus (SCN) of the hypothalamus of the brain. The SCN uses environmental signals, such as day length, food availability, and temperature, and adapts and aligns the circadian rhythm to these changes [1]. If any alterations occur in the circadian clock, then sleep patterns are disturbed, causing circadian rhythm sleep disorders (CRSDs) and ultimately leading to mutation and polymorphic changes in certain clock genes [2,3].

Drugs of abuse have been shown to have disruptive effects on the sleep-wake cycle and sleep quality, which eventually lead to the possibility of sleep disorders such as insomnia [4,5]. Additionally, insufficient sleep may also increase the risk of drug abuse, including cocaine addiction [6]. Changes in circadian rhythm mainly influence the brain’s ventral striatum and limbic forebrain regions [7]. In mammals, the circadian clock exists in nearly all cells of the body. Alterations in the circadian rhythm may be reflected in changes to the metabolism, breakdown and storage of fats and sugars, and the functions of individual metabolic pathways [7]. Moreover, cocaine abuse has been shown to cause both acute (short-term) and chronic (long-term) sleep dysfunction [8]. Cocaine abuse can negatively affect the brain pathways involved in maintaining circadian rhythms and may also influence drug use by triggering potential relapses [2,3]. The relationship between drugs of abuse and sleep disruption is bidirectional and complex. Sleep disruption leads to an increased risk of drug abuse, and conversely, drug abuse may cause CRSDs and has also been associated with neurodegenerative diseases [4,9,10]. Previous studies have reported that drug addiction or drug abuse is associated with an increased risk of heart, liver, lung, and brain disorders and psychological complications [11]. Neurobiological dysfunction connecting oscillations of the circadian rhythm with sleep disturbance has been linked to psychostimulant cocaine use-altered metabolic profiles [12,13,14].

Advances in metabolomics and analytical techniques have enabled the detection of hundreds of metabolites in body fluids and excreta that can be used to define biochemical profiles of drug abuse and addiction [4]. Metabolomics studies have been known to be used for the discovery of potential diagnostic, prognostic, and therapeutic biomarkers [15,16]. We utilized liquid chromatography–mass spectrometry (LC–MS) as a metabolomics analysis technique to investigate the effect of circadian rhythm sleep disruption with and without cocaine treatment on mouse plasma metabolite levels, which regulate various metabolic processes. Our investigations may also contribute insights into sleep disorders such as bruxism, thought to be mediated by neurological, dental, and genetic disorder components [17,18,19].

This study aimed to understand the impact of metabolite changes due to CRSD with and without cocaine exposure on metabolic processes. In the present work, we focused on identifying additional metabolic changes and altered metabolic pathways in cocaine-exposed CRSD for comparison of the differences with cocaine-exposed mice. To extend the coverage of the variable concentrations of metabolites, we chose an untargeted method based on plasma LC–MS/MS metabolite profiling. This multiplexed targeted LC–MS/MS approach has been shown to be quite robust and versatile in various biomarker and systems biology studies [16,20,21]. Several significant changes in various essential and energy metabolism-related metabolic pathways were observed in the present work. Utilizing multivariate statistical analysis, we combined the top-performing metabolite biomarkers into a model that distinguished between the control and cocaine-exposed groups and the CRSD with the cocaine-exposed group with excellent performance. Monitoring metabolites in cocaine-exposed CRSD mice with cocaine-exposed mice may improve the identification of metabolites that can act as specific biomarkers.

## 2. Materials and Methods

### 2.1. Animal Model

The 10- to 12-week-old C57BL/6J mice (both male and female procured from the Jackson Laboratory, Bar Harbor, Maine) used in the study were housed and acclimatized in polycarbonate cages at the vivarium. The optimum temperature of 23 °C was maintained under 12 h L/D photo cycles, and the mice were provided food and water ad libitum. Zeitgeber time 0 (ZT0) indicates the subjective start time of the day, and ZT12 is the subjective start time of the night under constant dark conditions over 24 h.

### 2.2. Treatment Schedule and Method

Animals were randomly assigned to 4 groups as follows: group 1: control (normal 12 h L/D cycle; group 2: circadian rhythm disruption (CRSD) (animals subjected to a 6 h phase advance every six days for eight cycles); group 3: cocaine (11 doses of 10 mg/kg/d, s.c. on alternate days for 22 days (12 h L/D cycle); group 4: CRSD with cocaine treatment (cocaine + CRSD) (circadian-disturbed animals treated with 11 doses of cocaine (10 mg/kg/d, s.c.) on alternate days for 22 days). Control animals were given saline alone (no drug). Five animals were assigned to each group (n = 5). The average weight of mice in each group was approximately 30 g. The body weight of each animal was checked weekly. We did not see any significant difference in weight changes between these groups (Appendix A). The final established optimum concentration and time response were used in subsequent experiments. Finally, the animals were decapitated to collect plasma for further analyses. All experiments were performed in compliance with protocols approved by the local committee for Animal Care and Texas A&M University Animal Use Committee and conducted according to the 2011 NIH Guide for the Care and Use of Laboratory Animals.

### 2.3. Circadian Rhythm Sleep Disruption Method

Animals were randomly assigned to 4 groups as follows: group 1: control (normal 12 h L/D cycle; group 2: circadian rhythm disruption (CRSD) (animals subjected to a 6 h phase advance every six days for eight cycles); group 3: cocaine (11 doses of 10 mg/kg/d, s.c. on alternate days for 22 days (12 h L/D cycle); group 4: CRSD with cocaine treatment (cocaine + CRSD). In order to disrupt the circadian rhythm in mice, five animals were subjected to a 6 h phase advance cycle before the end of normal 12 h L/D cycle. Once we subjected mice to phase advance cycle, we maintained these CRSD group mice in that phase for six days. After six days of that particular 12 h L/D cycle, we again subjected animals to another 6 h phase advance cycle before the end of previous 12 h L/D cycle. We followed this procedure every six days for eight cycles. We monitored circadian rhythm changes by recording wheel-running activities in CRSD group mice.

### 2.4. Plasma Sample Preparation

HPLC grade methanol (50 mL) was kept at −80 °C overnight. Then, the samples were centrifuged at 14,000× *g* for 10 min in a cold room (4–8 °C), and the supernatant was transferred to a new 1.5-mL microcentrifuge tube. We added enough methanol (cooled to −80 °C) to the supernatant to make a final 80% (vol/vol) methanol solution. For a 100 µL sample, 400 µL methanol was added, followed by vertexing for 1 min and incubating for 30 min at −80 °C. The samples were then centrifuged at 14,000× *g* for 10 min (4–8 °C). The supernatant was transferred to a new 1.5-mL microcentrifuge tube, and the samples were filtered through Target2™ PVDF syringe filters and then lyophilized to a pellet without heating. The dried metabolite samples can be stored at −80 °C for several weeks.

### 2.5. Mass Spectrometric Analyses

Mass spectrometric analyses were performed on a Sciex TripleTOF 6600 system (AB SCIEX, Framingham, MA, USA) equipped with an electrospray ionization (ESI) source used in positive and negative ionization modes. The ESI source conditions were as follows: nebulizer (Gas 1), 50 psi; heater (Gas 2), 45 psi; curtain gas flow, 30 psi; source temperature, 550 °C; ion spray voltage floating, +5500 V (+) and −4500 V (−).

Time-of-flight–mass spectrometry (TOF–MS) mode (full scan) and information-dependent acquisition (IDA) mode (product ion scan) were utilized for collecting MS and MS/MS data. For the TOF–MS scan, the mass range was from *m*/*z* 70 to 1000, and for the product ion scan, the mass range was from *m*/*z* 30 to 1000. The collision energy (CE) was set to 30 V (+) or −30 V (−), and the collision energy spread (CES) was ±15 V. The accumulation time was set to 0.25 sec for the TOF–MS scan and 0.05 sec for the product ion scan.

The MS instrument was automatically calibrated using a calibration delivery system injected with APCI positive and negative calibration solutions after every 5 samples. The system was controlled by Analyst TF 1.7.1 software (Sciex). Data were processed by Sciex OS software version 2.0.1. The metabolites were identified using their retention time and exact mass, and matching of the unknown MS/MS spectra with a spectrum from the standard MS/MS library led to the identification of 350 metabolites.

### 2.6. Chromatography Conditions

The mass spectrometer was coupled to a Shimadzu HPLC (Nexera X2 LC-30AD). Chromatography was performed under HILIC conditions using a SeQuant^®^ ZIC^®^-pHILIC 5 μm polymeric 150 × 2.1 mm PEEK-coated HPLC column from MilliporeSigma, USA. The column temperature, sample injection volume, and flow rate were set to 45 °C, 5 μL, and 0.15 mL/min, respectively. The HPLC conditions were as follows: Solvent A: 20 mM ammonium carbonate including 0.1% ammonium hydroxide, and Solvent B: acetonitrile. The gradient conditions were as follows: 0 min: 80% B, 20 min: 20% B, 21 min: 80% B, and 34 min: 80% B. The total run time was 34 min.

### 2.7. Statistical Analysis

Statistical analyses were carried out using GraphPad Prism 9 Statistical Software Package (GraphPad Software Inc., La Jolla, CA, USA). The interactive effect of either cocaine or cocaine in combination with circadian disruption compared with control animals was calculated using analysis of variance and Student’s *t* test as well as the nonparametric Mann–Whitney U test. Significance of comparisons between the control and cocaine exposure with CRSD is denoted by *p* < 0.05. We used the following equations to calculate metabolite concentration:Mass-to-charge ratio- *m*/*z* represents mass divided by charge number and the horizontal axis in a mass spectrum is expressed in units of *m*/*z*;(STDEV (five replicates of peak area for each metabolite)/AVERAGE (five replicates peak area for each metabolite)) ∗ 100.

## 3. Results

### 3.1. Extraction and Preprocessing

The detailed pictorial diagram depicts the overall workflow of the untargeted mass spectrometry (MS)-based metabolomics analysis that is focused on the global detection and relative quantification of metabolites in a biological sample, such as plasma, using the TripleTOF^®^ 6600 System from Sciex (Figure 1A–C). We utilized this approach for biomarker discovery and metabolic pathway perturbance identification caused by cocaine treatment and circadian rhythm changes. Blood samples were collected from the following treatment groups of 10- to 12-week-old mice (5 in each group): control, cocaine-treated, circadian wake-dependent modulated, and circadian wake-dependent modulated with cocaine treatment. The control and cocaine-treated mice were subjected to a normal 12 h light/12 h dark (12 h L/D) cycle. In contrast, mice with CRSD were subjected to a 6 h phase advance every six days for eight cycles, with or without cocaine treatment. Plasma samples were centrifuged and mixed with methanol to a final concentration of 80% (vol/vol). Finally, these solutions were lyophilized and processed for untargeted MS-based metabolomics analysis using the TripleTOF^®^ 6600 system (Figure 1A–C). The plasma polar and nonpolar metabolites in the control, cocaine, CRSD, and cocaine + CRSD samples were investigated (Figure 1A–C). The final dataset of 243 metabolites was identified across all sample groups based on their accurate mass and coelution with authentic metabolite standards, as shown in Table 1. We assessed the proportions of plasma metabolites in the targeted and untargeted matrices. Good instrumental stability was observed, as indicated by the 13–20% coefficient of variation (CV) values for the metabolites in the quality control samples. Untargeted analysis was performed in MRM mode and 243 (peaks (mz/rt)) data matrices, i.e., metabolites detected reliably in all plasma samples, were detected (Appendix A). We identified 279 metabolites in positive ion mode and 324 in negative ion mode. The implemented normalization procedures were grouped into four categories. Sample-specific normalization allowed us to manually adjust sample concentrations based on biological inputs. Row-wise normalization allowed general-purpose adjustment for differences among samples, and data transformation and scaling were two different approaches to make the features more comparable. Figure 1D shows the data distribution before and after normalization. Moreover, the plots and kernel densities for 50 metabolites after normalization are shown in Figure 2A.

### 3.2. Analysis

Fisher’s least significant difference method (Fisher’s LSD) and Tukey’s honestly significant difference (Tukey’s HSD) in MetaboAnalyst were used for univariate analyses, which provided a preliminary overview of the comparisons between the different levels that were significant given the *p* value threshold (Figure 2B). Principal component analysis (PCA) explains the directions that best describe the variance in a dataset (X) without referring to class labels (Y). The data were summarized into far fewer variables called scores, which were weighted averages of the original variables. Figure 2C shows the 2-D score plot between the selected principal components (PCs). To assess the significance of class discrimination, a permutation test was performed. In each permutation, a PLS-DA model was built between the data (X) and the permuted class label (Y) using the optimal number of components determined by model cross-validation based on the original class assignment.

There are two variable importance measures in PLS-DA. The first, variable importance in projection (VIP), is a weighted sum of the squares of the PLS loadings considering the amount of explained Y variation in each dimension. The other important measure is based on the weighted sum of PLS regression. Figure 2D shows the permutation test results for model validation, which identified important features by PLS-DA. The metabolite itaconate showed the highest VIP score among all groups (Figure 2D). Moreover, 2-hydroxyglutamate and citramalate also showed high VIP scores in all groups (Figure 2D). To maximize the sampling differences between the study groups and explore the metabolites most relevant to circadian rhythm and cocaine treatment, metabolites with a VIP > 1.5 were selected (Figure 2D). PLS-DA determined the VIP plot. The higher the VIP value is, the better the contribution of that metabolite to the separation of the groups. We compared the treatment groups with the control group to identify the significant metabolites expressed in the treatment groups for further analysis. We performed multiple *t* test analyses to visualize the data in a volcano plot by plotting the negative logarithm of the *p* value on the Y-axis (usually base 10). First, we compared the cocaine group with the control (Figure 3A). The volcano plot shows highly significant metabolites with a low *p* value at the top right (upregulated) and top left (downregulated). We observed that beta-glycerophosphate (BGP) and adenosine monophosphate (AMP) were significantly upregulated, while D-alanine (D-Ala) and O-acetylserine (OAS) were downregulated (Figure 3A). Moreover, when we compared CRSD to the control group, the volcano plot showed that trans-4-hydroxy-L-proline was significantly downregulated while AMP expression was significantly upregulated (Figure 3B). Interestingly, the volcano plot depicting the comparison between the control and cocaine + CRSD groups demonstrated that two metabolites, AMP, and inosine monophosphate (IMP), were significantly upregulated (Figure 3C). Furthermore, we compared the predominantly expressed metabolites from all the following three comparison groups: control vs. cocaine, control vs. CRSD, and control vs. cocaine + CRSD. The resulting analysis revealed 17 common metabolites, as shown in Figure 3D.

The metabolomic profiling of the plasma was performed, identifying metabolites based on their ionization efficiencies. We observed that some of these metabolites showed better sensitivity in negative ion mode than in positive ion mode. We identified 304 metabolites in negative ion mode while 279 metabolites in positive ion mode, as shown in Figure 4A (Appendix A). Next, to better characterize the changes in the circadian rhythm and cocaine treatment on the metabolome of mice, we compared the total peak intensity (the following scaling) of the identified metabolites across comparison groups (Appendix A). We observed that the cocaine + CRSD, CRSD, and cocaine groups showed lower total peak intensity compared to the control group (Figure 4B). This approach led to the detection and identification of a total of 243 metabolites across all samples. These samples belonged to mainly a superclass of Organic acids with 53% of total metabolites (Appendix A). Other metabolites belonged to various super classes such as 19% nucleic acids, 10% carbohydrates, 3% Benzenoids, 3% Glycerolipids, and 3% of fatty acyls (Figure 4C) (Appendix A). We further divided the metabolites into superclasses and arranged them based on their expression, such as upregulation and downregulation at each comparison with the control (cocaine + CRSD, CRSD, and cocaine groups) as shown in Figure 4D (see also Appendix A). Furthermore, we analyzed the subclasses of superclasses to which all these metabolites belong. We observed that these metabolites belong to subclasses of carbohydrates such as oligosaccharides, glycosyl compounds, and monosaccharides. Moreover, metabolites were classified into purines, pyrimidines, and nicotinamide subclasses of nucleic acids. Finally, a total of 80 metabolites were found to belong to subclasses including amino acids and peptides, TCA acids, and phosphate esters of the superclass of organic acids (Figure 4E).

Additionally, we obtained extracted ion chromatograms (XICs) for the metabolites that were significantly altered in cocaine-treated and CRSD-changed mouse plasma samples compared to the control; these chromatograms demonstrated good peak shapes. The metabolite inosine showed a peak area (PA) of 4.891 × 10^6^ with a retention time (RT) of 5.33 min in negative mode (Figure 5A). Similarly, we identified metabolites such as 10-hydroxydecanoate (negative mode; PA = 2.837 × 10^5^, RT = 1.86 min), adenosine 5′-diphosphate (positive mode; PA= 2.914 × 10^5^, RT = 9.23 min), uridine monophosphate (positive mode; PA = 3.524 × 10^5^, RT = 8.54 min), xanthine (positive mode; PA = 2.963 × 10^5^, RT = 6.95 min), and cortisol (positive mode; PA = 4.142 × 10^5^, RT = 1.89 min), as shown in Figure 5B–F. We utilized these 17 common metabolites for metabolite set enrichment analysis (MSEA), which was performed with the MetaboAnalyst tool. MSEA contains human, mammalian, and chemical class metabolite sets. This module accepts a list of compound names, compound names with concentrations, or a concentration table. The analysis is based on several libraries containing ~9000 biologically meaningful metabolite sets collected primarily from human studies, including >1500 chemical classes. The MSEA depicted the top 25 enriched pathways with significant *p* values and enrichment ratios. Cardiolipin biosynthesis, phosphatidylcholine biosynthesis, purine metabolism, pyrimidine metabolism, and methyl histidine metabolism were the most upregulated and distinct pathways among the 25 top pathways (Figure 6A). Similarly, the *p* values of the top 25 pathways were transformed into the negative logarithm of the *p* value to plot on the Y-axis (base 10) and were thus depicted in a logarithmic fashion (Figure 6B). We also investigated the significance and contribution of these metabolites to illustrate the impact of these pathways. We observed that purine and pyrimidine metabolism were the most impacted, with significant −log10(*p* values) (Figure 6C). Moreover, we calculated the average normalized peak areas for the replicates and each sample group and then compared these values using a heatmap. Figure 6D displays the heatmap, which shows the expression of the metabolites in each sample group. Metabolites such as inosine, hypoxanthine, xanthosine, NAD+, beta-nicotinamide, adenine dinucleotide, palmitoyl carnitine, 3-(2-hydroxyphenyl) propanoate, and glycerate showed higher expression in the cocaine and CRSD groups than in the control group. Intriguingly, 4-guanidinobutanoate, lauroyl carnitine, methionine, and palmitoyl carnitine expression were upregulated in the cocaine + CRSD group compared to the control. We utilized the 16 common metabolites for pathway analysis in the next analyses. The MetaboAnalyst tool performs metabolic pathway analysis (integrating pathway enrichment analysis and pathway topology analysis) and visualizes human and mouse genome pathways. Interestingly, this tool can also simultaneously analyze genes and metabolites of interest within the context of metabolic pathways. We used this tool to integrate evidence from cocaine exposure transcriptomics data into central nervous system (CNS) cells and metabolomics data from plasma samples. The resulting analysis revealed the metabolite-gene-disease interaction network (Figure 7A). This network showed significant interactions of metabolites, such as hippuric acid, with various diseases, including phenylketonuria, schizophrenia, propionic acidemia, and tyrosinemia type I. In-depth analysis of the metabolite-to-metabolite interactions revealed that N-acetyl galactosamine, glycerol-3-phosphate (G3P), cytidine monophosphate, and adenine were mainly affected by cocaine + CRSD treatment (Figure 7B). The final investigation revealed that metabolites such as N-acetyl galactosamine, G3P, cytidine monophosphate, and adenine mainly interacted with FAD, glycerol, NADH, citric acid, adenosine triphosphate (ATP), and uridine, which are primarily involved in energy metabolism processes (Figure 7B).

## 4. Discussion

Cocaine is a powerful psychostimulant, and users feel euphoric, energetic, and mentally alert after a short exposure time. However, the long-term effects of cocaine use include addiction, irritability and mood disturbances, restlessness, paranoia, and auditory hallucinations [22]. Cocaine use can further influence the brain’s reward system by disturbing the circadian clock and chemical transmission, leading to addiction. Cocaine overuse leads to heightened pleasure and addiction mediated by the release of dopamine in the brain. When euphoric, it is normal for the individual to experience sleep disturbances triggered by the drastic drug-induced energy spike.

In contrast, sleep-deprived individuals experience hypersomnia during withdrawal. These substance-induced sleep disturbances may harm their mood, resulting in relapse. Substance-induced insomnia can affect the body’s ability to heal and the mind’s ability to rejuvenate. Sleep and circadian rhythm are intricately connected to various hormonal and metabolic processes that maintain metabolic homeostasis. Research shows that sleep deprivation and sleep disorders may have profound metabolic and cardiovascular implications through multiple pathways involving sympathetic overstimulation, hormonal imbalance, and subclinical inflammation [23]. In parallel, the neurochemistry of drugs of abuse is different, and the question is whether each has a different effect on sleep.

Several clinical studies with various cohorts have reported that circadian rhythm disturbance can accelerate drug addiction [4]. Conversely, clinical studies have also demonstrated that illicit drug use is associated with sleep disorders and impairs both immune and neuronal functions [24]. Changes in the concentrations of metabolites are acute compared to changes in the expression levels of a gene or a protein, making measuring metabolites a relatively easy and sensitive method to determine the biological status. The changes in endogenous metabolites are attributed to the pathogenesis of several major diseases, as they seem to interrupt specific cellular pathways of the disease in question. Thus, metabolomics is crucial for deducing pathophysiological mechanisms and developing novel therapeutics [4,24,25]. The great advancements in technology and instrumentation have fostered the growth of nanoscale measurements of biological signatures present in urine, plasma, and blood [20]. Therefore, it is both imperative and informative to investigate the effects of cocaine treatment and changes in circadian rhythm at the metabolite level in plasma extracted from the blood of euthanized mice. A detailed characterization of the metabolic processes that are impacted by sleep disturbances is required to better understand the increased risk of the adverse metabolic effects observed in individuals that use drugs. Herein, we used an in vivo untargeted metabolomics approach to determine the impact of circadian rhythm changes and cocaine treatment on circulating metabolite levels in plasma.

The untargeted metabolomics approach identified 17 commonly significantly altered metabolites. Among these, uridine diphosphate, glycerol-2-phosphate, hippurate, G3P, uridine monophosphate, S-adenosylmethionine, AMP, BGP, and galactosamine were the main metabolites altered due to cocaine use and circadian rhythm changes. Uridine monophosphate and uridine diphosphate are esters of phosphoric acid with the nucleoside uridine, both of which were upregulated in cocaine-administered and circadian rhythm-changed mice [26]. In humans, the function of orotidylate decarboxylase is carried out by the protein UMP synthase. Defective UMP synthase can result in the metabolic disorder of orotic aciduria [27]. A previous in vivo model study demonstrated that uridine alters dopaminergic activity and the related behavioral impairments. The administration of uridine blunted the amphetamine-induced increase in striatal dopamine. These observations were interpreted as indications that chronic uridine modulates the stimulant-induced release of dopamine [28]. Additionally, another study showed that uridine administration for five consecutive days prevents REM sleep deprivation-induced deficits in learning and memory associated with neurocognitive function [29]. These previous reports suggest that sleep deprivation and neurochemical changes due to cocaine and circadian rhythm changes in mice may affect metabolic processes and induce uridine-associated metabolites to nullify the effects of the treatments. Metabolites such as S-adenosylmethionine AMP are derivatives of adenosine [30]. Adenosine is mainly derived from ATP via AMP and various enzymatic reactions [26]. Research studies have shown that cocaine increases the cerebral extracellular levels of adenosine, an endogenous purine nucleoside that modulates dopaminergic neurotransmission, which further induces adenosine A2A receptor (A2AR) stimulation [31,32,33,34]. Our metabolomics analysis showed that adenosine metabolites were significantly upregulated, pointing toward ATP-AMPK energy-associated pathways. Our previous research reports showed that cocaine influences energy metabolism in vitro and in vivo through epigenetic modulation [35,36,37]. G3P, glycerol-2-phosphate, and BGP are phosphoric esters of glycerol, which is a component of glycerophospholipids [38]. These metabolites are synthesized by reducing dihydroxyacetone phosphate (DHAP), a glycolysis intermediate, with G3P dehydrogenase. DHAP and thus G3P can also be synthesized from amino acids and citric acid cycle intermediates via the glyeroneogenesis pathway [38]. G3P dehydrogenases can be localized in the cytosol and the inner membrane of the mitochondria. G3P and DHAP can move across the mitochondrial outer membrane through porins and shuttle between two dehydrogenases due to their small size. This movement across membranes leads to NADH generation from cytosolic cellular mechanisms. For example, in glycolysis, DHAP is reduced to G3P due to NAD+ reoxidation, and the reducing counterpart generates a proton gradient across the inner membrane of the mitochondria via G3P oxidation and quinone reduction [39].

Moreover, our analysis demonstrated that these intermediate glycolysis metabolites were significantly altered in cocaine-treated mice and circadian rhythm-changed mice. Altered pathways included changes in cardiolipin biosynthesis, purine and pyrimidine metabolism, biosynthesis, degradation (glutamine, lysine, carnitine, methionine, and histidine), glycolysis (lactate and glucose), triglyceride biosynthesis, glycerol phosphate shuttle, and tricarboxylic acid (TCA) cycle (citrate cycle). Energy metabolism, the mitochondrial electron transport chain, and purine-pyrimidine metabolism dominate the altered biochemistry after cocaine treatment and circadian rhythm changes in mice. The accumulation of lactate and glucose, which is common in many diseases, comes with higher energy demands [40]. The increase in carnitine indicated increased carnitine, lysine, and glutamine biosynthesis activity connected with the TCA cycle via lactate accumulation, again in response to the higher energy demand [41].

Importantly, while it was beyond the scope of the current work, it remains a limitation of this study that we did not pursue further investigation of the identified metabolites in the untargeted mass spectrometry (MS)-based metabolomics analysis. Further investigation is needed to understand the involvement of purine and pyrimidine metabolites in the CRSD and cocaine-affected brain energy metabolism.

## 5. Conclusions and Future Direction

In conclusion, our results provide insight into the effects of chronic cocaine exposure and circadian rhythm changes on metabolic processes. This metabolomics and bioinformatics analysis is a comprehensive supplement that gives quick insights into how significant metabolite signature markers are regulated during cocaine exposure and circadian rhythm changes via the modulation of targeted metabolic pathways.

## Figures and Tables

**Figure 1 metabolites-12-00869-f001:**
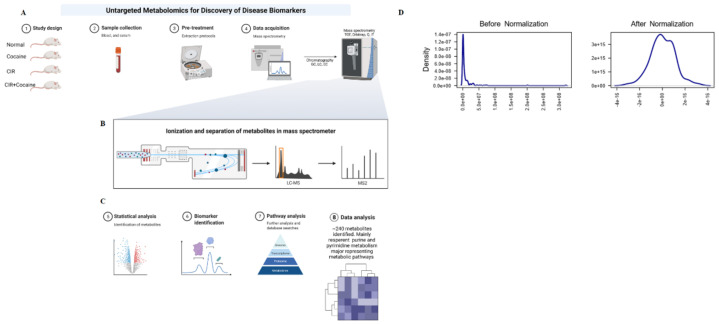
Workflow of the metabolomic data analysis of the plasma samples. (**A**–**C**) Schematic representation of the untargeted metabolome data workflow regarding quantification, annotation, and identification of metabolites in the control, circadian disruption (CRSD), cocaine, and CRSD with cocaine treatment groups. (**D**) Metabolite data analysis before and after normalization.

**Figure 2 metabolites-12-00869-f002:**
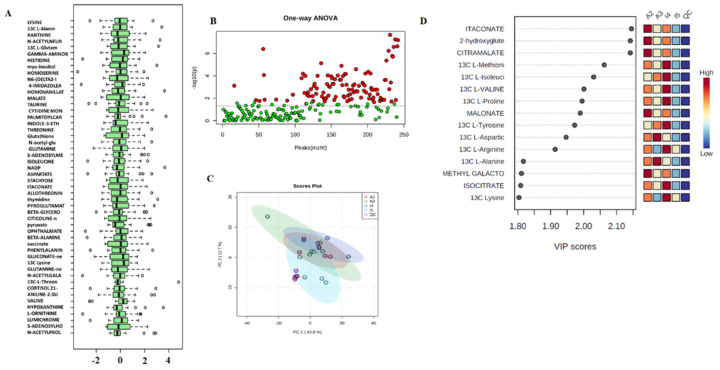
Data processing and normalization the metabolomic data from plasma samples. (**A**) Boxplots and kernel density plots before and after normalization. The boxplots show at most 50 features due to space limitations. The density plots are based on all samples. Selected methods: row-wise normalization: N/A; data transformation: N/A; data scaling: autoscaling. (**B**) Important features selected by ANOVA with a *p* value threshold 0.05. (**C**) Score plot between the selected PCs. The explained variances are shown in brackets. (**D**) Important features identified by PLS-DA. The colored boxes on the right indicate the relative concentrations of the corresponding metabolites in each group under study.

**Figure 3 metabolites-12-00869-f003:**
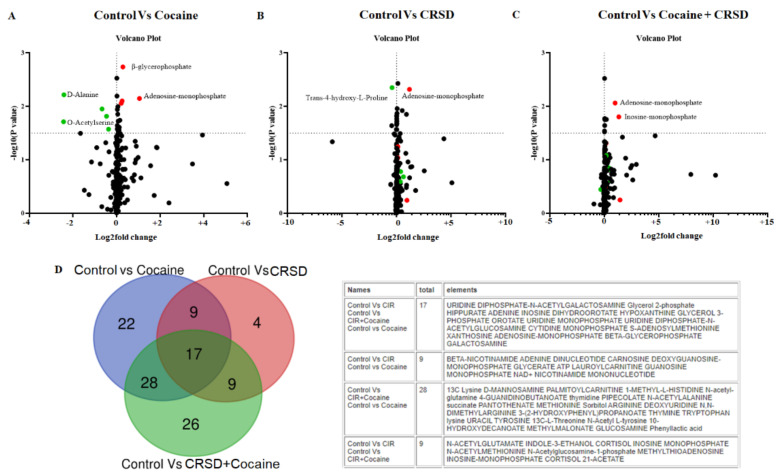
Summary of metabolite changes in the mouse plasma samples of the circadian disruption (CRSD), cocaine, CRSD, and control groups. (**A**) Volcano plot presenting the differential expression of metabolites in the cocaine group compared to the control. The X-axis corresponds to fold changes (FCs) of −2 (downregulation) and +2 (upregulation). The Y-axis represents the −log10(*p* value). The red and green points on the plot represent the significantly differentially expressed (DE) metabolites. (**B**) Volcano plot presenting the differential expression of metabolites in the CRSD group compared to the control. The X-axis corresponds to FCs of −2 (downregulation) and +2 (upregulation). The Y-axis represents the −log10(*p* value). The red and green points in the plot represent the significantly DE metabolites. (**C**) Volcano plot presenting the differential expression of metabolites in the cocaine + CRSD group compared to the control. The X-axis corresponds to FCs of −2 (downregulation) and +2 (upregulation). The Y-axis represents the −log10(*p* value). The red and green points in the plot represent the significantly DE metabolites. (**D**) Venn diagram showing the 17 common identified metabolites among the control vs. cocaine, control vs. CRSD and control vs. cocaine + CRSD groups.

**Figure 4 metabolites-12-00869-f004:**
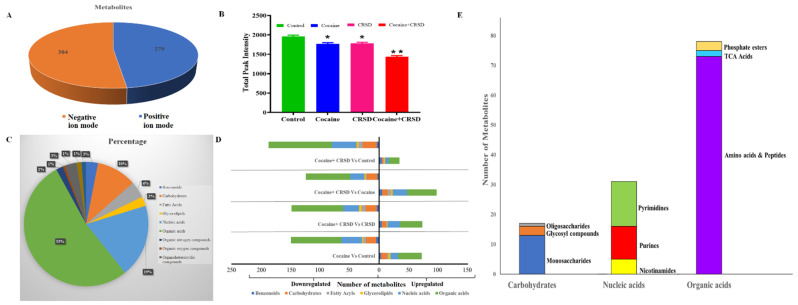
Overview of effect of cocaine and CRSD on metabolomic changes. (**A**) Pie chart showing the total of 304 metabolites and 279 metabolites identified by using positive and negative ion mode of untargeted liquid chromatography-tandem mass spectroscopy, respectively. (**B**) The bar diagram representing the total peak intensities across control, cocaine + CRSD, CRSD, and cocaine groups. (**C**) The pie chart representing the percentage of metabolites belong to superclass of metabolites. (**D**) The stack bar diagram representing the total number of metabolites upregulated and downregulated at each comparison groups (Cocaine vs. Control, Cocaine + CRSD vs. Control, Cocaine + CRSD vs. Cocaine, and Cocaine + CRSD vs. CRSD) and distribution of the metabolites per superclass of metabolites. (**E**) The stack bar diagram representing the number of metabolites divided on the basis of subclasses under the superclass. * and ** represent the levels of significance at 10% and 5%, respectively.

**Figure 5 metabolites-12-00869-f005:**
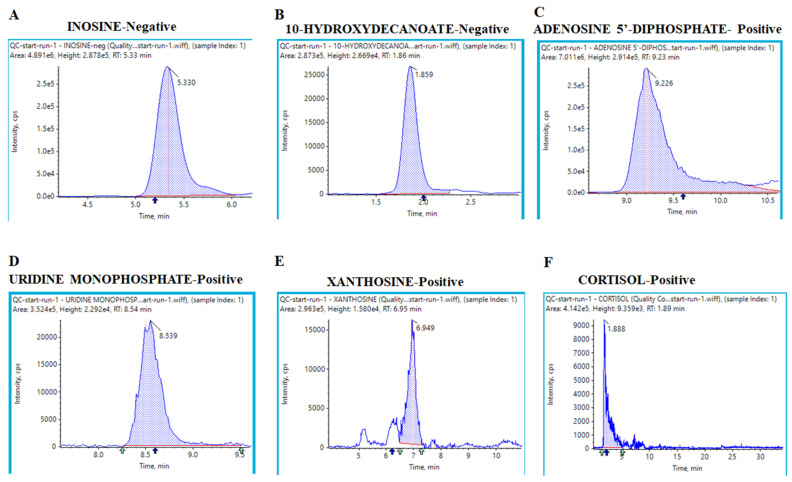
Extracted ion chromatograms of the significantly expressed metabolites. Extracted ion chromatograms from LC–MS/MS analysis of the plasma metabolites (blue line) and their deuterated internal standards (red line). The data were obtained from a pooled plasma sample. The chromatograms demonstrate clear retention time and peak area overlap with the deuterated internal standards.

**Figure 6 metabolites-12-00869-f006:**
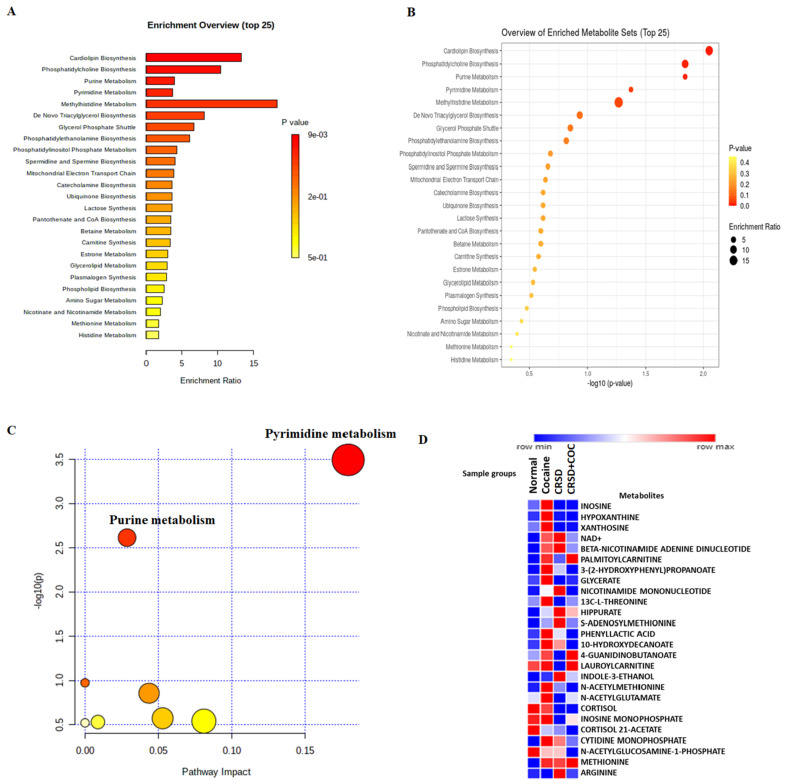
Functional analysis of the common significantly altered metabolites. (**A**) The involvement of metabolites in various pathways were investigated using enrichment pathway analysis. A list of the most enriched pathways in which these metabolites are involved is shown based on the enrichment ratio. (**B**) The involvement of metabolites in various pathways was investigated using enrichment pathway analysis. A list of the most enriched pathways in which these metabolites are involved is shown based on −log10(*p* value). (**C**) The pathway impact is shown based on −log10(*p* value). (**D**) Heatmap showing the top upregulated metabolites (red) and downregulated metabolites (blue) across the control, circadian disruption (CRSD), cocaine, and CRSD with cocaine treatment groups.

**Figure 7 metabolites-12-00869-f007:**
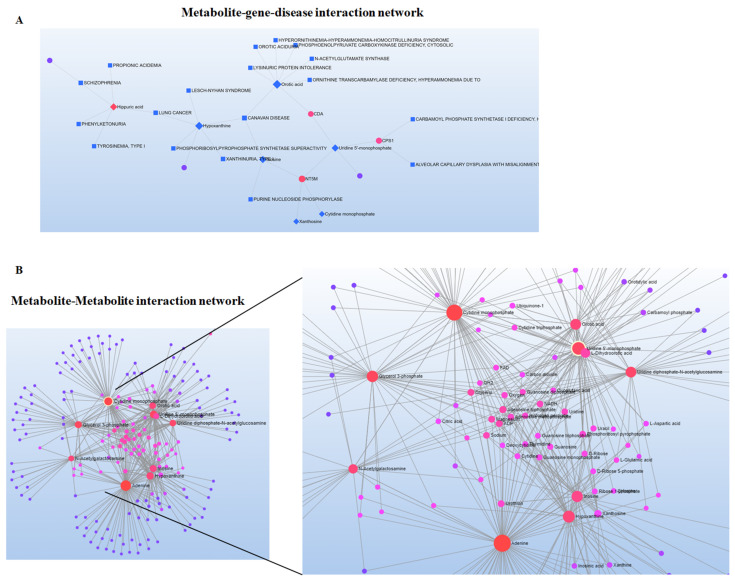
Metabolite interaction network. (**A**) Metabolite-gene-disease interaction network analysis of the annotated significantly differentially accumulated metabolites. The square blue boxes represent diseases, and significant metabolites are shown in red. (**B**) Metabolite–metabolite interaction network analysis of the annotated significantly differentially accumulated metabolites. The metabolites are represented as nodes (circles), and enzymes are represented as edges (lines). Metabolites with higher importance are highlighted in red followed by pink and blue. The size of the node indicates the node degree (the number of links a particular node has to other nodes).

**Table 1 metabolites-12-00869-t001:** Summary of data processing results for all samples.

Samples	Features (Positive)	Missing/Zero	Features (Processed)
A2-CIR-1	243	0	242
A2-CIR-2	243	0	242
A2-CIR-3	243	0	242
A2-CIR-4	243	0	242
A2-CIR-5	243	0	242
A3-COC-CIR-1	243	0	242
A3-COC-CIR-2	243	0	242
A3-COC-CIR-3	243	0	242
A3-COC-CIR-4	243	0	242
A3-COC-CIR-5	243	0	242
I4-Control-1	243	0	242
I4-Control-2	243	0	242
I4-Control-3	243	0	242
I4-Control-4	243	0	242
I4-Control-5	243	0	242
I5-COCAINE-1	243	0	242
I5-COCAINE-2	243	0	242
I5-COCAINE-3	243	0	242
I5-COCAIN-4	243	0	242
I5-COCAIN-5	243	0	242
QC-start-run-1	243	0	242
QC-start-run-2	243	0	242
QC-start-run-3	243	0	242
QC-end-run-1	243	0	242
QC-end-run-2	243	0	242
QC-end-run-3	243	0	242

## Data Availability

The authors confirm that the data supporting the findings of this study are available within the article and/or its Appendix A.

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
