# Peer review of "Sleep Disorder and Cocaine Abuse Impact Purine and Pyrimidine Nucleotide Metabolic Signatures"

_metabolites, 2022, doi:10.3390/metabo12090869_

Round 1

Reviewer 1 Report

 The authors have submitted a manuscript assessing the effect of circadian rhythm sleep disruption with and without cocaine treatment on mouse plasma metabolite levels, which regulate various metabolic processes. The study aimed to understand the impact of metabolite changes due to CRSD with and without cocaine exposure on metabolic processes. The authors conclude that their results provide insight into the effects of chronic cocaine exposure and circadian rhythm changes on metabolic processes. This study is addressing an important issue given high prevalence of CRSD worldwide and its critical role for the vulnerability of individuals to drug abuse.

The manuscript is well written. Methods and description are generally sound. Statistical methods used are comprehensive and relevant.

There are a few minor issues that should be addressed:

1. In the ‘Materials and methods’ section, it would be better to describe in more details how were CRSD group animals subjected to the phase advancement protocol, procedures duration, how did you prove the presence of CRSD in these group?

2. In the ‘Materials and methods’ section please indicate the body weight of the experimental animals.

3. In the ‘Materials and methods’ section no information regarding the cocaine dosage and administration route/ self-administration!

4. Page 2. Line 87, please replace the “Circadian time” with “Zeitgeber time” (ZT)

5. page 1, line 12, line 41 please replace “sleep cycle” with “sleep-wake cycle”.

6. Figure 1. please, replace “CID”  with “CRSD”

7. Fig.1,  C7- the text visibility is very low after zooming, please correct.

Author Response

Review-1:  The manuscript is well written. Methods and descriptions are generally sound. The statistical methods used are comprehensive and relevant.

There are a few minor issues that should be addressed:

  1. In the ‘Materials and methods section, it would be better to describe in more detail how were CRSD group animals subjected to the phase advancement protocol, the procedures duration, and how did you prove the presence of CRSD in these group?

Ans- Thank you for your suggestions. We have added sub-section of “2.3. Circadian rhythm sleep disruption method” under the Materials and Methods section section. Please check page 3, lines 110-121 (highlighted in green).

  1. In the ‘Materials and methods section, please indicate the body weight of the experimental animals.

Ans- We have added the body weight of the experimental animals. Please check page 3, lines 102-104 (highlighted in green). We have given the experimental animals' body weights in supplementary table 1.

  1. In the ‘Materials and methods section no information regarding the cocaine dosage and administration route/ self-administration!

Ans- Apologies for this omission.  As requested, we have added the dose and route of administration (10 mg/kg/d, s.c.) for cocaine.  Please find this information on pages 2 and 3, lines 96-97, 98-99, and 112-113 (highlighted in green).

  1. Page 2. Line 87, please replace the “Circadian time” with “Zeitgeber time” (ZT)

Ans- Thank you for your suggestion. We have replaced the “Circadian time” with “Zeitgeber time” (ZT). Please check Page 2, lines 92-93. The sentence is highlighted in green color for clarity.

  1. Page 1, line 12, line 41, please replace “sleep cycle” with “sleep-wake cycle”.

Ans- Thank you for your suggestion. We have replaced the “sleep cycle” with “sleep-wake cycle” on pages 1, line 12, and line 42. The sentence is highlighted in green color.

  1. Figure 1. please, replace “CID”  with “CRSD”

Ans- Thank you for the suggestions. We have replaced “CID”  with “CRSD” in Figure 1.

  1. Fig.1,  C7- the text visibility is very low after zooming; please correct it.

Ans- We have updated all figures' quality. Thank you for your suggestions.

Reviewer 2 Report

Please find the attachment, i am recommending for major revision of this study. 

Author Response

Review-2:

  1. The authors should improve the quality of all figures.

Ans- We have updated all figures quality. Thank you for your suggestions.

  1. The authors should rewrite the title.

Ans- Thank you for suggestions. We have rewritten the manuscript title, focusing on the key findings.

  1. The authors should add one section about “Related Work” after the introduction. In this section, the authors should add the work related to sleep disorders such as bruxism (# A novel hybrid machine learning classification for the detection of bruxism patients using physiological signals, Appl. Sci. 10 (2020) 1–16. https://doi.org/10.3390/app10217410. # Sleep Bruxism Detection Using Decision Tree Method by the Combination of C4-P4 and C4-A1 Channels of Scalp EEG, IEEE Access. 7 (2019) 102542–102553. https://doi.org/10.1109/ACCESS.2019.2928020. # Detection, Treatment Planning, and Genetic Predisposition of Bruxism: A Systematic Mapping Process and Network Visualization Technique, CNS Neurol. Disord. - Drug Targets. 20 (2020) 755–775. https://doi.org/10.2174/1871527319666201110124954), Insomnia (# Progress in Detection of Insomnia Sleep Disorder: A Comprehensive Review, Curr. Drug Targets. 22 (2020) 672–684. https://doi.org/10.2174/1389450121666201027125828. # M.B. Bin Heyat, Insomnia: Medical Sleep Disorder & Diagnosis, 1st ed., Anchor Academic Publishing, Hamburg, Germany, 2016. https://www.anchorpublishing.com/document/337729.), Sleep Apnea, Narcolepsy, etc.

Ans- Thank you for the suggestion. While bruxism was not measured in this investigation, we acknowledge our findings may touch on this issue.  Accordingly, we have added one section about “Related Work” after the introduction. Please check Page 2 lines 83-85. The new content is highlighted in red color.

  1. Authors should write the mathematical expression of the Statistical Methods.

Ans- We have added the mathematical expression in the Statistical Methods. Please check Page 4 line 167-170. The sentences are highlighted in red color.

  1. Authors should divide the results into sub-sections for eg, preprocessing, extraction, etc.

Ans- As per reviewers suggested, we have divided the results in sub-sections 3.1 preprocessing and extraction and 3.2 Analysis. Please check Page 4, line 172, Page 6 line 219. The changes are now highlighted in red color.

  1. Please add the limitations and Applications sub-section under discussion. Also, add the conclusion section separately with Future Directions.

Ans- Thank you for the suggestions; we have added the limitations and Applications sub-section under discussion. Also, add the conclusion section separately with Future Directions. Please check Page 14 lines 466-478. The sentences are highlighted in red color.

Round 2

Reviewer 2 Report

Dear Authors, 

   I appreciate your efforts. But I have some minor suggestions: 

1. Please remove related work. Add these sentences (Our investigations may also contribute insights into sleep disorders such as bruxism, 82 thought to be mediated by neurological, dental, and genetic disorder components [21–23].) after metabolic processes. (line no-65)

2. Please change the paragraph and then write "This study aimed to understand the impact of metabolite changes 65 due to CRSD with and without cocaine exposure on metabolic processes" and combine the below paragraph with this new paragraph. 

3.  please start the new paragraph where you mentioned the limitations of the study *(line no- 465). 

Author Response

We appreciate the editors' decision to require minor revisions and have done so in the resubmitted draft. Responding to the reviewer's comments:

1) Please remove related work. "Add these sentences (Our investigations may also contribute insights into sleep disorders such as bruxism, thought to be mediated by neurological, dental, and genetic disorder components [21–23].) after metabolic processes. (line no-65) "

We have moved the section of text (requested by Reviewer 2 in original reviews) to line 65 in the introduction as requested.

  1. Please change the paragraph and then write, "This study aimed to understand the impact of metabolite changes due to CRSD with and without cocaine exposure on metabolic processes" and combine the below paragraph with this new paragraph.  

We have combined the purpose statement with the last paragraph of the introduction section, as requested.

  1. Please start the new paragraph where you mentioned the study's limitations *(line no- 465).  

…The limitations we described in the discussion section have now been isolated in their own paragraph (at line 464) as requested by the reviewer.